**Mid-Pliocene Atlantic Meridional Overturning Circulation simulated in PlioMIP2**

Zhongshi Zhang[1], Xiangyu Li[1], Chuncheng Guo[2], Odd Helge Otterå[2,3], Kerim H. Nisancioglu[4], Ning Tan[5], Camille Contoux[6], Gilles Ramstein[6], Ran Feng[7], Bette L. Otto-Bliesner[8], Esther Brady[8], Deepak Chandan[9], W. Richard Peltier[9], Michiel L. J. Baatsen[10], Anna S. von der Heydt[10], Julia E. Weiffenbach[10], Christian Stepanek[11], Gerrit Lohmann[11,12], Qiong Zhang[13], Qiang Li[13], Mark A. Chandler[14], Linda E. Sohl[14], Alan M. Haywood[15], Stephen J. Hunter[15], Julia C. Tindall[15],Charles Williams[16], Daniel J. Lunt[16], Wing-Le Chan[17], Ayako Abe-Ouchi[17]

1. Department of Atmospheric Science, School of Environmental studies, China University of Geoscience, Wuhan, 430074, China
2. NORCE Norwegian Research Centre, Bjerknes Centre for Climate Research, 5007 Bergen, Norway
3. Center for Early Sapiens Behaviour, 5007 Bergen, Norway
4. Department of Earth Science and Bjerknes Centre for Climate Research, University of Bergen, 5007 Bergen, Norway
5. Key Laboratory of Cenozoic Geology and Environment, Institute of Geology and Geophysics, Chinese Academy of Sciences, Beijing 100029, China
6. Laboratoire des Sciences du Climat et de l'Environnement, LSCE/IPSL, CEA-CNRS-UVSQ, Université Paris-Saclay, F-91191 Gif-sur-Yvette, France
7. Department of Geosciences, University of Connecticut, Storrs, USA
8. Climate and Global Dynamics Laboratory, National Center for Atmospheric Research, Boulder, USA
9. Department of Physics, University of Toronto, Toronto, Canada
10.Institute for Marine and Atmospheric research Utrecht (IMAU), Department of Physics, Utrecht University, Utrecht, The Netherlands.
11.Alfred Wegener Institute – Helmholtz Centre for Polar and Marine Research, Bremerhaven, Germany
12.Institute for Environmental Physics, University of Bremen, Bremen, Germany
13. Department of Physical Geography and Bolin Centre for Climate Research, Stockholm University, Stockholm, Sweden
14. CCSR/GISS, Columbia University, New York, USA
15. School of Earth and Environment, University of Leeds, Woodhouse Lane, Leeds, West Yorkshire, LS29JT, UK
16. School of Geographical Sciences, University of Bristol, Bristol, UK.
17. Atmosphere and Ocean Research Institute (AORI), University of Tokyo, Kashiwa, Japan

Correspondence: Zhongshi Zhang(zhongshi.zhang@cug.edu.cn)

**Abstract**

In the Pliocene Model Intercomparison Project phase 2 (PlioMIP2), coupled climate models have been used to simulate an interglacial climate during the mid-Piacenzian warm period (mPWP, 3.264 to 3.025 Ma). Here, we compare the Atlantic Meridional Overturning Circulation (AMOC), poleward ocean heat transport and sea surface warming in the Atlantic simulated with these models. In PlioMIP2, all models simulate an intensified mid-Pliocene AMOC. However, there is no consistent response in the simulated Atlantic ocean heat transport, or the depth of the Atlantic overturning cell. The models show a large spread in the simulated AMOC maximum, the Atlantic ocean heat transport, as well as the surface warming in the North Atlantic. Although a few models simulate a surface warming of ~8–12 °C in the North Atlantic, similar to the reconstruction from Pliocene Research, Interpretation and Synoptic Mapping (PRISM) version 4, most models appear to underestimate this warming. The large model-spread and model-data discrepancies in the PlioMIP2 ensemble does not support the hypothesis that an intensification of the AMOC, together with an increase in northward ocean heat transport, is the dominant mechanism for the mid-Pliocene warm climate over the North Atlantic.

**1. Introduction**

The mid-Piacenzian warm period (mPWP, 3.264–3.025 Ma) was a recent period of sustained warmth in geological history, with land-sea distribution, topography and levels of greenhouse gases being comparable to today (Dowsett et al., 2010, 2016; Haywood et al., 2010, 2016a). The estimated global mean temperature during the mPWP was 2–4°C higher than the pre-industrial (e.g., Dowsett et al., 2010, 2016; Haywood et al., 2010, 2016a), and the atmospheric $CO_2$ level was above 400ppmv (Badger et al., 2013). Thus, the mPWP climate is often thought of as a plausible test case that has the potential to provide insights for our future climate (e.g., Zubakov and Borzenkova, 1988; Haywood et al., 2016b; Burke et al., 2018).

To understand the mPWP climate, the Pliocene Modelling Intercomparison Project (PlioMIP) phase 1 was launched in 2010 (Haywood et al., 2010). The major forcing considered in PlioMIP1 was an increase (compared to pre-industrial) in the atmospheric $CO_2$ level to 405 ppmv, combined with a modern land-sea distribution (Haywood et al., 2013). The PlioMIP1 simulations (e.g., Chan et al., 2011; Bragg et al., 2012; Contoux et al., 2012; Kamae and Ueda, 2012; Stepanek and Lohmann, 2012; Zhang et al., 2012; Chandler et al., 2013; Rosenbloom et al., 2013) showed that the global annual mean surface air temperature (SAT) was 1.9–3.6°C warmer than pre-industrial in the multi-model ensemble mean (Haywood et al., 2013), while the

strength of Atlantic Meridional Overturning Circulation (AMOC) was similar to the pre-industrial level (Zhang et al., 2013a). However, when compared to marine (Dowsett et al., 2012, 2013) and terrestrial reconstructions (Salzmann et al. 2013), there was a large model-data discrepancy (Haywood et al., 2013) in the North Atlantic and the land realm of the Northern Hemisphere. The PlioMIP1 simulated surface warming in the North Atlantic is ~4–6°C smaller than the reconstruction. Because the PlioMIP1 simulations (Zhang et al., 2013a, 2013b) did not support a stronger Pliocene AMOC (compared to preindustrial) and an inferred enhancement of Atlantic northward ocean heat transport (OHT) suggested by proxies (Dowsett, 1992; Raymo et al., 1996), it was difficult to explain the reconstructed strong surface warming in the high-latitude North Atlantic during the mid-Pliocene.

To further understand the mPWP climate and to improve upon the model-data discrepancy, the PlioMIP phase 2 was initiated (Haywood et al., 2016a). PlioMIP2 employs the state-of-the-art boundary conditions from the Pliocene Research, Interpretation and Synoptic Mapping (PRISM) version 4 (Dowsett et al., 2016a), and focuses on the KM5c interglacial period (3.205 Ma) during the mPWP (Haywood et al., 2016a). The PRISM4 boundary conditions include reconstructed ocean bathymetry and land–ice surface topography, and also incorporate Pliocene soils and lakes (Dowsett et al., 2016; Haywood et al., 2016a). The most important change in boundary conditions in the northern high latitudes is the closure of the Arctic gateways, including the Canadian Archipelago and the Bering Strait (Haywood et al., 2016a). In PlioMIP2, the simulated global annual mean SAT increases by 1.7–5.2°C relative to the pre-industrial, with a multi-model mean SAT increase of 3.2°C (Haywood et al., 2020). In the Arctic, the simulated annual mean SAT increases by 3.7–11.6 °C compared to the pre-industrial, with a multi-model mean increase of 7.2 °C (de Nooijer et al., 2020).

In this study, we investigate the simulated AMOC in PlioMIP2, in order to further address the question whether an intensified AMOC and enhanced Atlantic OHT can explain the reconstructed North Atlantic-Arctic sea surface warming during the mPWP. In section 2, we briefly introduce the models that participated in PlioMIP2. In section 3, we compare the simulated AMOC and Atlantic OHT between PlioMIP1 and PlioMIP2. In section 4, we investigate the relationship between the simulated AMOC response and changes in North Atlantic SST. Finally, the results are discussed and summarized in section 5.

## 2. Introduction of models used in PlioMIP2

In this study, we analyze simulations with the fifteen models that have participated and provided the simulated AMOC results to PlioMIP2 (Table 1). All fifteen models have performed simulations according to the PlioMIP2 experimental protocol (Haywood et al., 2016). They provide the pre-industrial control

experiment (*pi-E280*) and the mid-Pliocene experiment (*midPliocene-Eoi400*) as a minimum. In the mid-Pliocene experiment, a land-sea mask with the Arctic gateways closed and an atmospheric $CO_2$ level of 400 ppmv are used. The atmospheric $CO_2$ level is in line with the very latest high-resolution proxy reconstruction based on Boron isotopes for ~3.2 Ma (Chalk et al. 2018). More details on the individual models and experimental design are introduced in a recent synthesis study (Haywood et al., 2020) and several individual modeling studies (Chandan and Peltier, 2017; 2018; Hunter et al., 2019; Chan and Abe-Ouchi, 2020; Döscher et al., 2020; Feng et al., 2020; Li et al., 2020; Lurton et al., 2020; Stepanek et al., 2020; Tan et al., 2020). In addition to these fifteen models, MRI-CGCM (Kamae et al., 2016) and HadGEM3-GC31-LL have taken part in PlioMIP2. However, MRI-CGCM and HadGEM3-GC31-LL are not considered in detail here, because MRI-CGCM did not provide the AMOC results to the PlioMIP2 database, and HadGEM3-GC31-LL did not use the enhanced land-sea distribution condition with the Arctic gateways closed instead using the modern land-sea distribution. Note five models come from the CCSM/CESM family in the PlioMIP2 ensemble. To avoid these models taking undue weights in the PlioMIP2 ensemble, median instead of mean values are used in this study.

Of the fifteen PlioMIP2 models used here, six of them also took part in PlioMIP1. They are CCSM4, COSMOS, HadCM3, IPSL-CM5A-LR, MIROC4m and NorESM-L. However, all these six models have submitted new pre-industrial control experiments to the PlioMIP2 database. CCSM4 has also been employed in a modified form by other modelling groups and referred to herein as CCSM4-UoT and CCSM4-Utrecht. Therefore, the pre-industrial AMOC maximums and depths in PlioMIP2 are slightly different to the values in PlioMIP1.

## 3. Simulated AMOC and OHT

### 3.1 Simulated AMOC in PlioMIP2

The PlioMIP2 models produce reasonable simulations for the pre-industrial AMOC. The pre-industrial modelled AMOC maximums (the maximum of the Atlantic meridional overturning streamfunction) range from ~10 to 28 Sv (1 Sv = $10^6$ $m^3$ $s^{-1}$; Table 1, Fig. 1). The multi-model median value of the AMOC maximums is 19.8 Sv, which is comparable to the observational AMOC strength of 18.7 ± 2.1 Sv at 26.5°N (Kanzow et al. 2010). The depths of the Atlantic overturning cell range from 2300 m to 3800 m.

In PlioMIP2, the models show that the maximum AMOC is enhanced by 1% to 53% in the mid-Pliocene, relative to the pre-industrial (Table 1, Fig. 1). The median value of the enhancement in maximum AMOC is 19%. Seven models (CCSM-UoT, COSMOS, GISS-E2-1-G, HadCM3,

IPSL-CM5A-LR, IPSL-CM5A2-LR, IPSL-CM6A-LR) show small changes in the mean depth of AMOC cell (the mean depth of positive streamfunction) in the mid-Pliocene (with depth changes of less than 100 m), when compared to the pre-industrial. However, five models, CCSM4, CESM1.2, CESM2, EC-Earth3-LR and MIROC4m, simulate a shoaling of the Atlantic overturning cell for the mid-Pliocene, with a shoaling of ~1190m, ~1330m, ~820m, ~350 m and ~440 m. On the other hand, three models, CCSM4-Utrecht, NorESM1-F, and NorESM-L, simulate a deeper mid-Pliocene Atlantic overturning cell with increases in the depth of ~540m, ~1590 m and ~1330 m (Fig. 1, 2).

Compared to PlioMIP1 (Zhang et al., 2013a), the simulated AMOC responses to Pliocene boundary conditions are different in PlioMIP2 (Fig. 2). In PlioMIP1, there was no consistent increase in the maximum strength of the AMOC, while there was a consistent shoaling of the Atlantic overturning cell. However, in PlioMIP2, there is a consistent increase in the maximum strength of the AMOC, while there is no consistent change in the depth of Atlantic overturning cell.

### 3.2 Simulated Atlantic OHT in PlioMIP2

As expected from the intensified AMOC, most models simulate an enhanced Atlantic OHT (averaged between 30°S and 80°N) in the mid-Pliocene experiments relative to the pre-industrial (Table 1, Fig. 3). The increases range from 4% to 39%. The largest enhancement is found in the simulation with IPSL-CM5A2-LR, while the smallest one is simulated with NorESM1-F. In contrast, six models, CCSM4, CESM1.2, CESM2, GISS-E2-1-G, MIROC4m and NorESM-L show a decrease (ranging from -1% to -17%) in Atlantic OHT.

Obviously, there is no linear relationship between the intensification in AMOC and the changes in mean Atlantic OHT in the PlioMIP2 simulations (Fig. 2b). For example, GISS-E2-1-G and IPSL-CM6A-LR both simulate increases of 24% in the AMOC maximum. However, GISS-E2-1-G shows a decrease in mean Atlantic OHT by -1%, while IPSL-CM6A-LR shows an increase of 29%. CCSM4 and CCSM4-Utrecht also show the same increase of 11% in the AMOC maximum, but opposite responses in the mean Atlantic OHT. This large model-spread in PlioMIP2 suggests that the relationship between AMOC strength and Atlantic northward OHT are highly model-dependent.

### 4. Simulated North Atlantic sea surface warming

In PlioMIP2, the simulated mid-Pliocene global annual mean SST is between 1.2 and 4.0 °C warmer than the pre-industrial. Most models show that the strongest sea surface warming appears in the mid-to-high latitude North Atlantic (Fig. 4, 5). The median of multi-model ensemble shows that SST increases by ~2-8 °C

in the North Atlantic between 30ºN and 80ºN (Fig. 6). The largest increase in ensemble median by 6-8 ℃ appears in the Labrador Sea south of Cape Farewell (the southernmost point of Greenland). EC-Earth3-LR simulates the largest increase in the North Atlantic SST above 12℃ in the mid-Pliocene experiment (Fig. 4).

However, the SST increases in the North Atlantic (averaged between 30ºN and 80ºN) in response to the changes in AMOC maximum and North Atlantic OHT (averaged between 30ºN and 80ºN) are highly model-dependent (Fig. 5). Of the fifteen PlioMIP2 models, eleven models simulate a mean SST increase between 2 and 4 ℃ in the North Atlantic. The ranges of the changes in AMOC maximum (from 1% to 53%) and mean North Atlantic OHT (from -13% to 43%) are large. Meanwhile, EC-Earth3-LR produces an increase of ~8 ℃ in mean North Atlantic SST, which is associated with an intensification of 3.2 Sv (19%) in the AMOC maximum and an enhancement of 0.16 PW (41%) in the mean North Atlantic OHT. CCSM4-UoT, CCSM4-Utrecht, and CESM2 produce a similar increase of ~5 ℃ in the mean North Atlantic SST, while the intensification in AMOC maximum shows a large range covering 0.9 Sv (4%), 2.1 Sv (11%), and 4.7 Sv (21%), whereas the mean North Atlantic OHT changes by 0.06 PW (9%), 0.04 PW (6%), -0.02 PW (-4%).

In PlioMIP2, the surface warming simulated with CCSM4-UoT, CCSM4-Utrecht, CESM2 and EC-Earth3-LR is close to or warmer than the PRISM4 reconstructions (Foley and Dowsett, 2019) in the North Atlantic between 30ºN and 80ºN, whereas the other models still appear to underestimate the North Atlantic SST (Fig. 6). A previous study (Otto-Bliesner et al., 2017) showed that the closing of the Arctic gateways led to warmer North Atlantic SSTs in the mid-Pliocene experiment, when compared to the pre-industrial. However, in all PlioMIP2 simulations analyzed here the Arctic gateways are closed, but not all of them simulate the warm North Atlantic SSTs as reconstructed in the PRISM4 data set (Foley and Dowsett, 2019). Although the Arctic gateways may lead to a better agreement between simulated and reconstructed mid-Pliocene North Atlantic SSTs in some models, the effect is either not present for all of the models or it is not of sufficient amplitude to fully resolve the model-data discord. The PlioMIP2 models show a larger model-spread in the simulated mid-Pliocene SST increases in the high-latitude North Atlantic, as well as the responses in AMOC and North Atlantic OHT, relative to PlioMIP1. This reduced agreement is not surprising as the model spread in global average surface temperatures is likewise more pronounced in PlioMIP2 (1.86–3.60 °C in PlioMIP1 (Haywood et al., 2013) compared to 1.7–5.2 °C in PlioMIP2 (Haywood et al., 2020).

**5. Discussion and summary**

Compared to the PlioMIP1 ensemble in which the Arctic gateways were kept open, all PlioMIP2 models forced with the PRISM4 reconstructions that consider the closed Arctic gateways simulate an intensification in the mid-Pliocene AMOC. CCSM4, COSMOS, HadCM3, IPSL-CM5A-LR, MIROC4m and NorESM-L have all participated in both PlioMIP1 and PlioMIP 2. These six models simulate an increase (compared to the pre-industrial) in the mid-Pliocene AMOC maximum that is larger in PlioMIP2 than in PlioMIP1, supporting the hypothesis that closed Arctic gateways is a requirement for the intensification of the mid-Pliocene AMOC. There are several further lines of evidence that support this hypothesis. HadGEM3-GC31-LL, which carried out the mid-Pliocene experiment forced with the PlioMIP2 boundary conditions, except with the land-sea distribution condition identical to the pre-industrial, produces a weaker mid-Pliocene AMOC (with a maximum of 14.3 Sv) compared to the pre-industrial (with a maximum of 16.1 Sv). With COSMOS, a sensitivity experiment forced with the modern land-sea distribution (the Arctic gateways opened) also shows a weaker AMOC, when compared to the core mid-Pliocene simulation (Stepanek et al., 2020). As revealed in the earlier study (Otto-Bliesner et al., 2017), the closed Arctic gateways lead to a stronger AMOC by inhibiting Arctic freshwater export to the North Atlantic. However, the magnitude of intensification in AMOC due to the closed Arctic gateways seems highly model-dependent. Some simulations suggest that the AMOC is enhanced by ~2 Sv due to the closed Bering Strait (Brierley and Fedorov, 2016; Otto-Bliesner et al., 2017), while some unpublished simulations in PlioMIP2 show much larger responses. Without consistent sensitivity experiments for the Arctic gateways, it remains difficult to reveal the range of model-spread on the gateways' impacts in PlioMIP2. This model-dependence will be addressed in more dedicated sensitivity experiments in the future.

In PlioMIP2, the large-model spread does not support the notion that an intensified mid-Pliocene AMOC is the principal mechanism responsible for the simulated warming of the North Atlantic SSTs. Compared to CCSM4, both CCSM4-UoT and CCSM4-Utrecht (Table 1) simulate warmer SSTs in the North Atlantic, suggesting that the increased background ocean vertical mixing parameters likely contribute to the strong mid-Pliocene North Atlantic warming simulated with these two models. Each model's climate sensitivity also influences the simulated mid-Pliocene warming in PlioMIP2. For example, relative to CCSM4 and CESM1.2, CESM2 has a greater equilibrium climate sensitivity (Feng et al., 2020; Haywood et al., 2020) and simulates the strongest North Atlantic warming in the mid-Pliocene experiment. With the modern land-sea distribution conditions, HadGEM3-GC31-LL simulates a weakened mid-Pliocene AMOC, but very warmer SSTs in the North Atlantic as well as an increase in the mid-Pliocene global mean SST (SAT) of 3.8°C (5.1°C) relative to the pre-industrial, which is the second largest warming in PlioMIP2 (Fig.

4). Moreover, a new lake and soil condition is employed in PlioMIP2 (Haywood et al., 2016). Methods for modifying the soil condition and their impacts on climate in the models are highly model-dependent, due to the large variety of land surface schemes included in the PlioMIP2 models, which could further amplify the diversity of warming signals in high latitude regions. Since not all models carried out the sensitivity experiments designed in PlioMIP2, it remains difficult to distinguish which change in boundary conditions is more dominant for the strong mid-Pliocene North Atlantic surface warming. Earlier studies (e.g., Feng et al., 2017) have noticed that the North Atlantic warming is not a unique feature in many mid-Pliocene simulations, since the warming in the North Pacific is also remarkable (Fig 4). This inter-basin symmetry suggests a potentially important component of the zonal mean polar amplification of the SST warming across the North Atlantic. Energy balance analyses (Hill, 2014; Feng et al., 2017) show that amplified zonal mean northern high latitude warming is dominated by regional radiative feedbacks from lowered surface albedo and enhanced high latitude greenhouse effect (from changes in water vapor), even with an enhanced AMOC by gateway closure.

It should be noted that observations of strong high-latitude warming in the North Atlantic are not sufficient to constrain the strength of AMOC or OHT (Zhang et al., 2013b). The AMOC strength measures the contrast in water transport between the upper and lower branches of the Atlantic cells, but the OHT is also influenced by the contrast in water temperature as well as the depth of AMOC. Moreover, OHT can be decomposed into a (vertical) MOC component and a (horizontal) gyre component. While the MOC component dominates in most of the Atlantic region, the gyre component has a comparable magnitude in the subpolar region (Williams et al., 2015). Therefore, there is no one-to-one correspondence between AMOC and OHT, especially in the subpolar regions. Furthermore, the SST warming pattern is not entirely determined by OHT, as demonstrated by the simulations both in PlioMIP1 and PlioMIP2.

Nevertheless, the PlioMIP2 experiments simulate a sea surface warming that is in better agreement with the PRISM4 reconstructions (Foley and Dowsett, 2019) in the North Atlantic, relative to the PlioMIP1 ensemble. As shown in the synthesis paper by Haywood et al. (2020), the multi-model means (with equal weight for each model) agree well with the reconstructions at the North Atlantic Sites 609, 1308, and show only small differences from the reconstructions at Sites 982, 642. The comparison between the PlioMIP2 simulations and the SST reconstructions in the KM5c interglacial (McClymont et al., 2020) also demonstrates the reduced model-data discord.

However, the improved model-data agreement in the North Atlantic is primarily caused by the relatively warm mid-Pliocene simulations run with EC-Earth3-LR and the five models from the

CCSM/CESM family (Fig. 6). For the other models, the range of warming at these sites is similar to that of PlioMIP1. This large model-spread suggests that the reconstructed strong mid-Pliocene sea surface warming in the North Atlantic is not necessarily caused by the intensified AMOC and enhanced Atlantic northward OHT as suggested previously (Dowsett, 1992; Raymo et al., 1996). Even given the intensified AMOC in PlioMIP2 due to the closed Arctic gateways, most models produce the mid-Pliocene North Atlantic sea surface warming weaker than the PRISM4 reconstruction (Foley and Dowsett, 2019).

Although the model-data discrepancy is reduced in the North Atlantic partly due to the intensified AMOC, the model-data mismatch remains large in other regions in PlioMIP2, for example Sites 1081, 1082, 1084, 1087 in the Benguela upwelling region (Fig. 6). The PRISM4 (Foley and Dowsett, 2019) and other syntheses of Pliocene SST (Fedorov et al., 2013, McClymont et al., 2020) reconstruct that the SSTs are about 6–8 °C warmer than today in the Benguela upwelling region. All PlioMIP2 models underestimate this warming in the PlioMIP2 (Fig. 6). Even EC-Earth3-LR, which produces the warmest mid-Pliocene simulation in the North Atlantic, only simulates 2–4 °C sea surface warming in the Benguela upwelling region.

A major feature of the mid-Pliocene seems to be the large increase in SST (about 2–10 °C) in the mid-latitude coastal upwelling regions and the relatively smaller increases in SST (about 2–4 °C) in the mid- to high latitudes (Fedorov et al., 2013) compared to the pre-industrial, though some studies suggest that SST reconstructions in upwelling regions are highly proxy-dependent (e.g., Leduc et al., 2014). For example, in the Benguela upwelling region, the Mg/Ca-based SST is colder than the alkenone-based SST by ~3-10 °C (Leduc et al., 2014). In the California upwelling region, Foley and Dowsett (2019) show that the Pliocene SST is similar to today, whereas Fedorov et al. (2013) show the regional SST is about 2-8 °C warmer than today. Despite the uncertainties in reconstructions, the simulated warming in the mid-latitude upwelling regions in PlioMIP2 can be found in the low end of the proxy-estimated range. Realistic simulations in upwelling regions require good model-abilities in simulating large-scale ocean stratification and sea surface wind stress (Miller and Tziperman, 2017; Li et al., 2019), which are partly model-resolution dependent in both atmosphere and ocean models (Gent et al., 2010; Small et al., 2015).

Taken together, these model-data discrepancies make it difficult to associate the intensified AMOC and enhanced Atlantic northward OHT with the reconstructed high mid-Pliocene SSTs. Fedorov et al. (2013) have suggested a possible mechanism for understanding the warm SSTs during the mPWP. Increased mixing in the subtropical ocean and reduced extratropical cloud albedo cause a strong warming in the mid-latitudes, including some upwelling regions. In PlioMIP2, CCSM4-UoT and CCSM4-Utrecht have considered

increasing the ocean background mixing parameters, but no model has tested the impact of a reduction of the extratropical cloud albedo in the mid-Pliocene experiments. This mechanism can be further addressed in future to investigate whether it is a suitable candidate for improving the simulation for upwelling regions.

Furthermore, it remains problematic to use the intensified AMOC to explain other features of the mid-Pliocene ocean circulation. During the mPWP, the vertical and meridional $\delta^{13}C$ gradients are reduced in the Atlantic. This can be explained with the increased ventilation in the Southern Ocean and does not necessarily depend on an intensified AMOC (Zhang et al., 2013b). However, simulations of Southern Ocean dynamics are highly model-dependent (Zhang et al., 2013a). In addition to the Southern Ocean, the Pliocene deep ocean circulation in the North Pacific appears different to the present day. In the subarctic North Pacific, high accumulation rates of calcium carbonate and biogenic opal suggest a strong deep convection there, thus the existence of North Pacific deep-water formation and a Pacific meridional overturning circulation (PMOC, Burls et al., 2017). However, with an intensified AMOC, a PMOC remains absent in the PlioMIP2 simulations.

In summary, all fifteen coupled models in PlioMIP2 used in this study simulate an intensified mid-Pliocene AMOC, relative to the pre-industrial. The simulated AMOC maximum (the maximum of the Atlantic meridional overturning streamfunction) increases by between 1% to 53%. However, these models do not simulate a consistent change in the depth of the Atlantic overturning cell and the Atlantic OHT. The spread in the responses of AMOC and Atlantic OHT in the models becomes larger in PlioMIP2, when compared to PlioMIP1. In the North Atlantic, EC-Earth3-LR and the models from the CCSM/CESM family can simulate an SST increase (~8–12 °C) close to the PRISM4 reconstruction, while other models appear to underestimate the sea surface warming. In PlioMIP2, the model-data discrepancy is reduced in the North Atlantic, but the discrepancy remains large in the upwelling regions. The large model-spread and the remaining model-data discrepancy suggests that an intensified AMOC and an enhanced Atlantic northward OHT cannot explain the reconstructed warm climate of the mid-Pliocene surface oceans.

**Data availability**

Complete data for PlioMIP2 is available upon request from Alan M. Haywood (a.m.haywood@leeds.ac.uk) to access the PlioMIP2 database. PlioMIP2 data from CESM2, EC-Earth3-LR, GISS-E2-1-G, IPSL-CM6A-LR and NorESM1-F can be obtained from the Earth System Grid Federation (ESGF) (https://esgf-node.llnl.gov/search/cmip6/, last access:3 December 2020, ESGF, 2020).

**Author contributions**

Z.Z. and X.L. analysed the data and wrote the draft of the paper. All authors contributed to discussion of the results and writing of the paper.

323

**Competing interests**

324

The authors declare that they have no conflict of interest.

326

**Special issue statement**

327

This article is part of the special issues "PlioMIP Phase 2: experimental design, implementation and scien-tific results" and "Paleoclimate Modelling Intercomparison Project phase 4 (PMIP4) (CP/GMD inter-journal SI)". It is not associated with a conference.

331

**Acknowledgements**

ZZ, XL were supported by the National Natural Science Foundation of China (Grant No. 41888101), the National Key Research and Development Program of China (Grant No. 2018YFA0605602), the China Scholarship Council (Grant no. 201804910023), the China Postdoctoral Science Foundation (Grant no. 2015M581154), the Norwegian Research Council (Project No. 221712, 229819, and 262618), the NordForsk-funded project GREENICE (Project No. 61841), as well as computing resources from Notur/Norstore projects NN9133/NS9133, NN9486/NS9486.

RF, BLO-B, and ECB acknowledge the CESM project, which is supported primarily by the National Science Foundation (NSF). This material is based upon work supported by the National Center for Atmospheric Research (NCAR), which is a major facility sponsored by the NSF under Cooperative Agreement No. 1852977. This research was additionally sponsored by U.S. National Science Foundation Grants 1903650 and 1814029 to RF and 1418411 to BLO-B. Computing and data storage resources, including the Cheyenne supercomputer (doi:10.5065/D6RX99HX), were provided by the Computational and Information Systems Laboratory (CISL) at NCAR.

WRP and DC were supported by Canadian NSERC Discovery Grant A9627 and they wish to acknowledge the support of SciNet HPC Consortium for providing computing facilities. SciNet is funded by the Canada Foundation for Innovation under the auspices of Compute Canada, the Government of Ontario, the Ontario Research Fund – Research Excellence, and the University of Toronto.

AvdH and MLJB, acknowledge the program of the Netherlands Earth System Science Centre (NESSC), financially supported by the Ministry of Education, Culture and Science (OCW, grant #. 024.002.001). Simulations with CCSM4-Utrecht were performed at the SURFsara dutch national computing facilities and were sponsored by NWO-EW (Netherlands Organisation for Scientific Research, Exact Sciences) under the project 17189.

GL and CS acknowledge use of computational resources from the Computing and Data Centre of the Alfred-Wegener-Institute – Helmholtz-Centre for Polar and Marine Research towards generation of the COSMOS PlioMIP2 simulation ensemble. GL acknowledges funding via the Alfred Wegener Institute's research programme PACES2. CS acknowledges funding by the Helmholtz Climate Initiative REKLIM and the Alfred Wegener Institute's research programme PACES2.

QZ acknowledges the financial support by the Swedish Research Council (Vetenskapsrådet, grant no. 2013-06476 and 2017-04232). The model simulations with EC-Earth3 and data analysis were performed by resources provided by ECMWF's computing and archive facilities and the Swedish National Infrastructure for Computing (SNIC) at the National Supercomputer Centre (NSC) partially funded by the Swedish Research Council through grant agreement no. 2016-07213.

AMH, SJH and JCT, acknowledge the FP7 Ideas: European Research Council (grant no. PLIO-ESS, 278636), the Past Earth Network (EPSRC grant no. EP/M008.363/1) and the University of Leeds Advanced Research Computing service. JCT was also supported through the Centre for Environmental Modelling And Computation (CEMAC), University of Leeds.

WLC and AAO acknowledge funding from JSPS KAKENHI grant 17H06104 and MEXT KAKENHI grant 17H06323, and JAMSTEC for use of the Earth Simulator supercomputer.

365

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

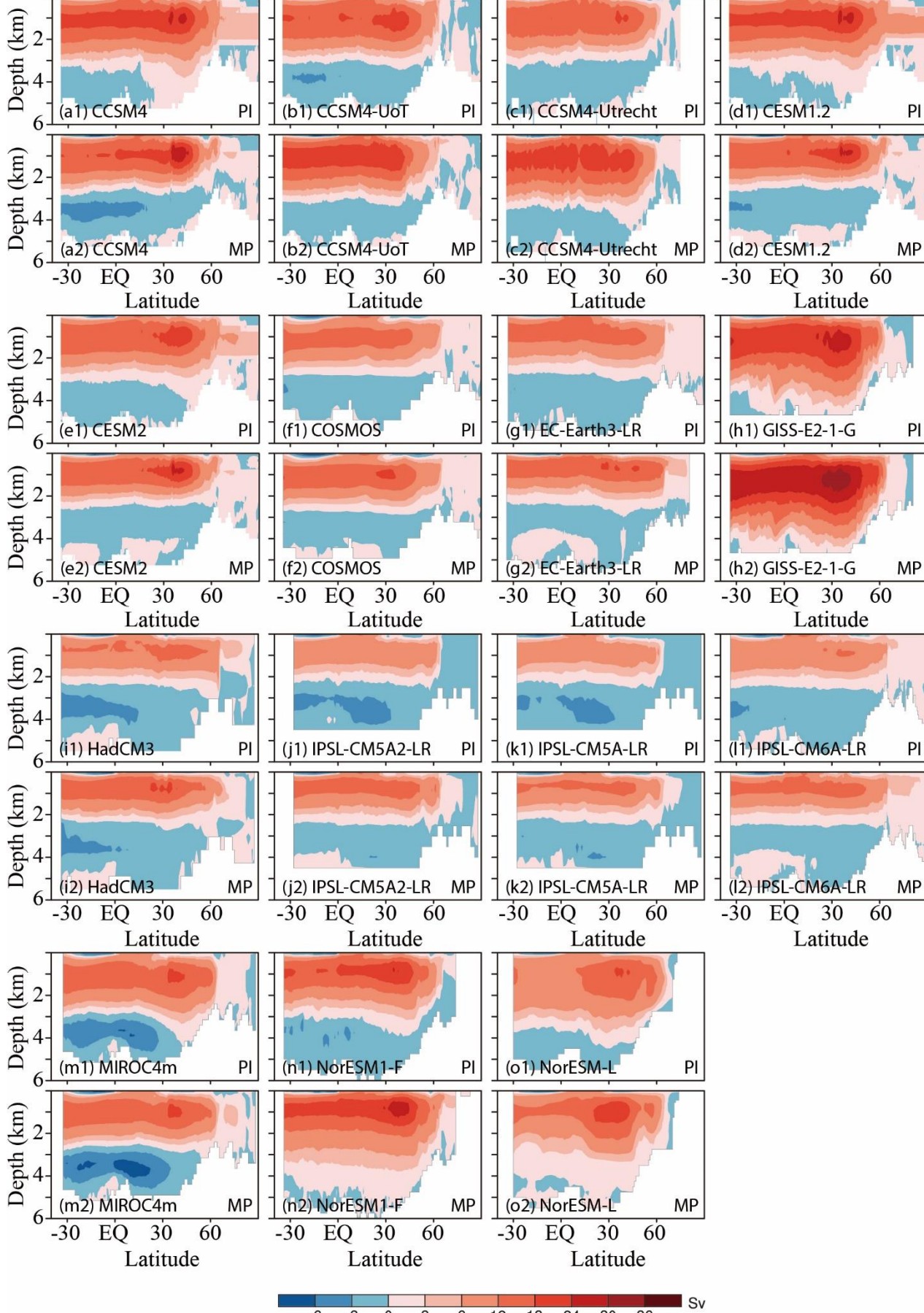

**Fig. 1. The simulated AMOC (unit: Sv) in PlioMIP2.** PI means the pre-industrial. MP means the mid-Pliocene.

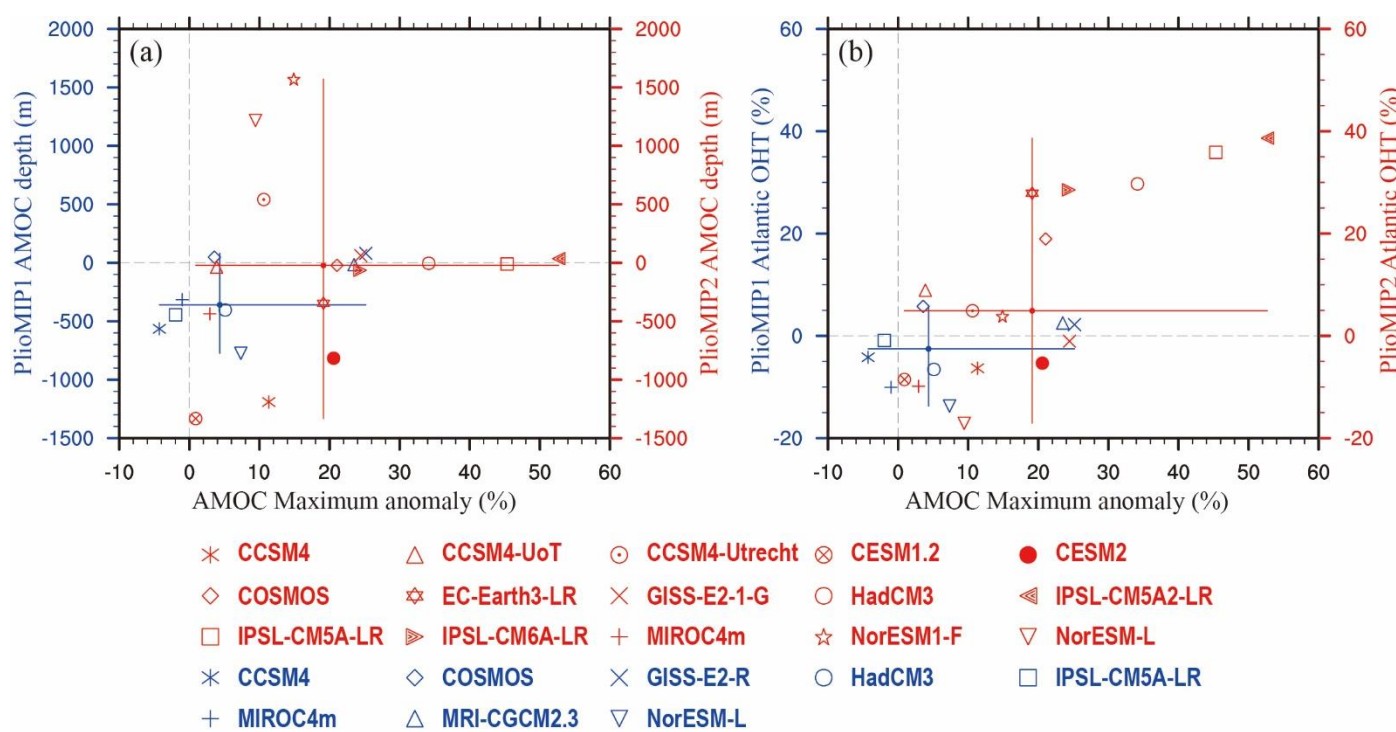

**Fig. 2. Simulated changes in AMOC maximum, depth and Atlantic northward OHT.** (a) Changes in AMOC maximum (unit: %) vs. responses in the mean depth of AMOC cell (unit: m). (b) Changes in AMOC maximum (unit: %) vs. responses in the mean ocean heat transport in Atlantic between 30 °S and 80 °N (unit: %). The blue markers show the PlioMIP1 simulations. The red markers show the PlioMIP2 simulations. The vertical and horizontal lines show the model range, while the intersection of these lines indicates the median value. Note only the mean values of AMOC maximum, depth and Atlantic northward OHT for each model are used here to calculate the anomalies, significant tests based on time serials are not employed.

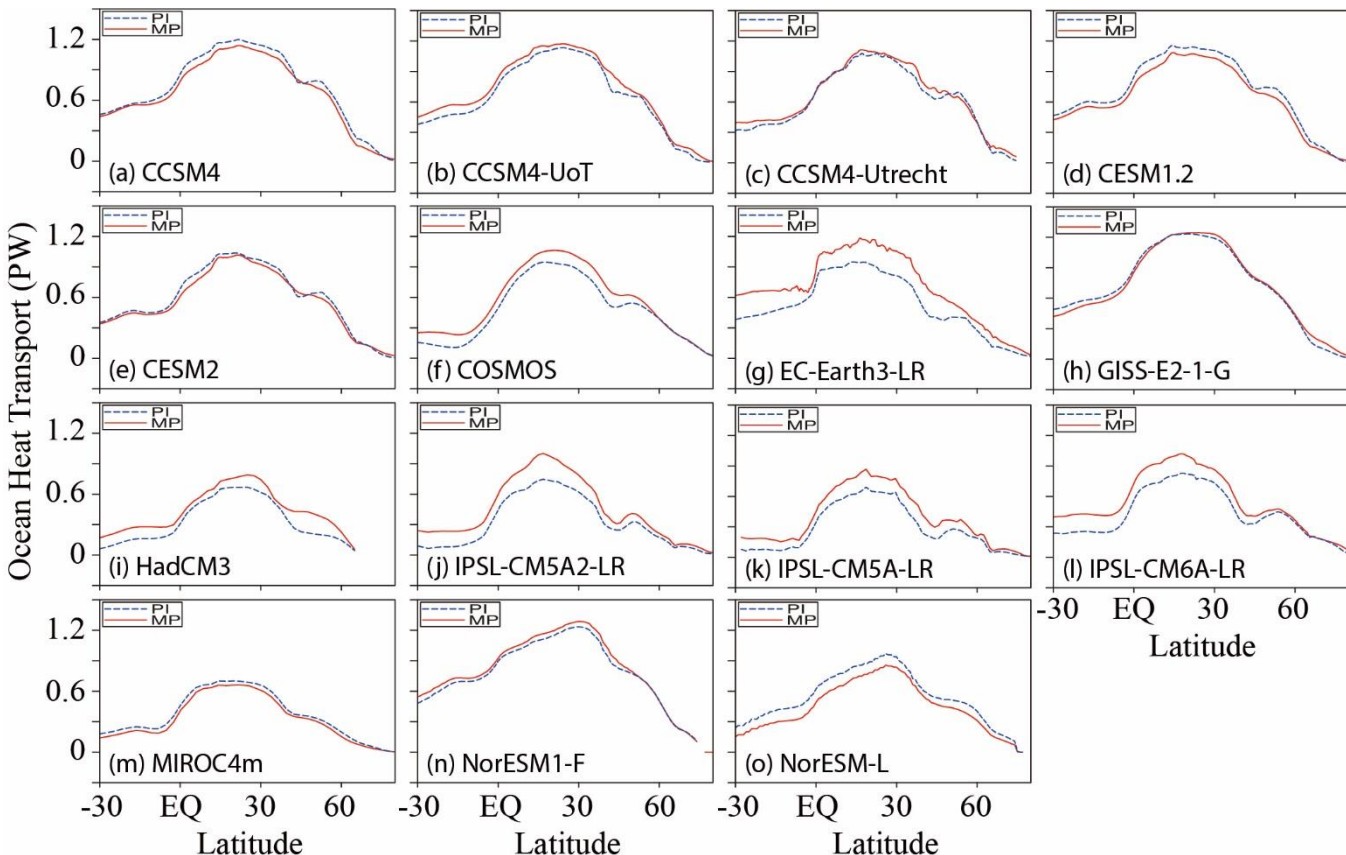

**Fig. 3. Simulated Atlantic poleward oceanic heat transport in the PlioMIP2 (unit: PW).** Blue dashed lines show the pre-industrial, and red solid lines show the mid-Pliocene.

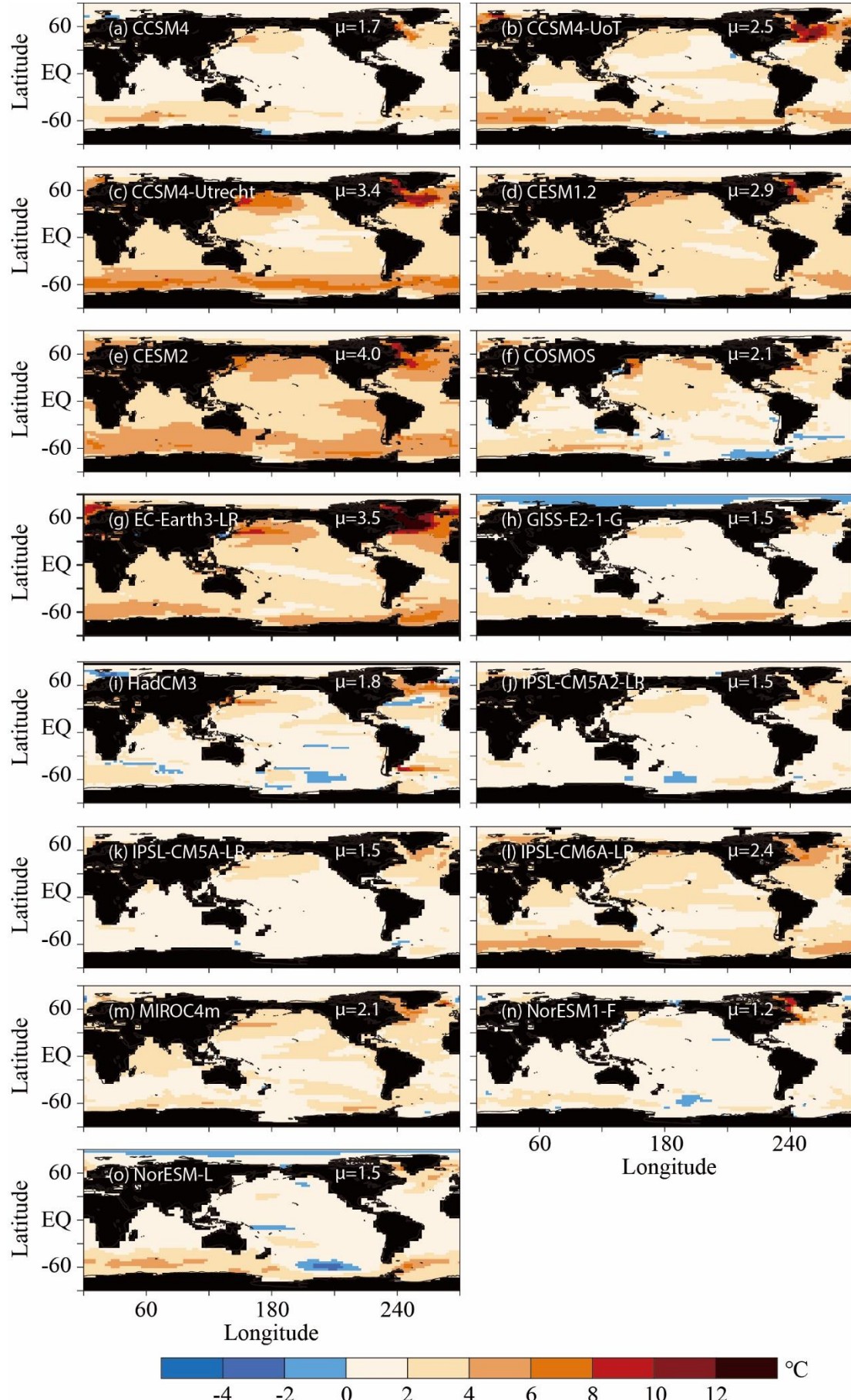

**Fig. 4. Simulated mid-Pliocene annual SST anomalies in PlioMIP2 (units: °C).** μ means the global mean.

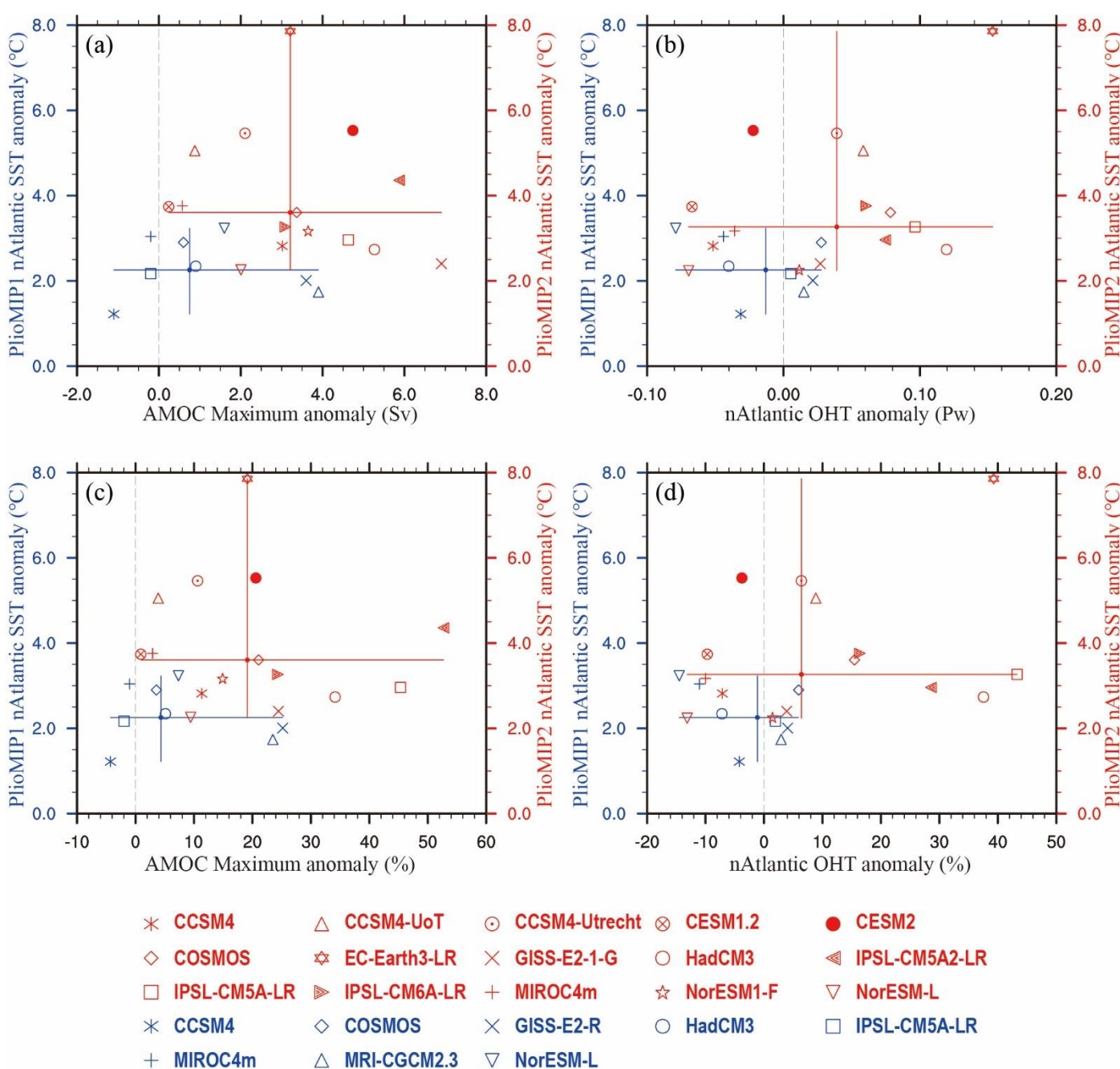

**Fig. 5. Simulated changes in AMOC maximum, North Atlantic OHT, and responses in high-latitude North Atlantic SST.** The North Atlantic OHT is the averaged value between 30 °N and 80 °N (unit: Pw). The high-latitude North Atlantic includes the Atlantic and Greenland-Iceland-Norwegian (GIN) seas between 30 °N and 80 °N. Note only the mean values of AMOC maximum, North Atlantic OHT and SST for each model are used to calculate the anomalies, significant tests based on time serials are not employed.

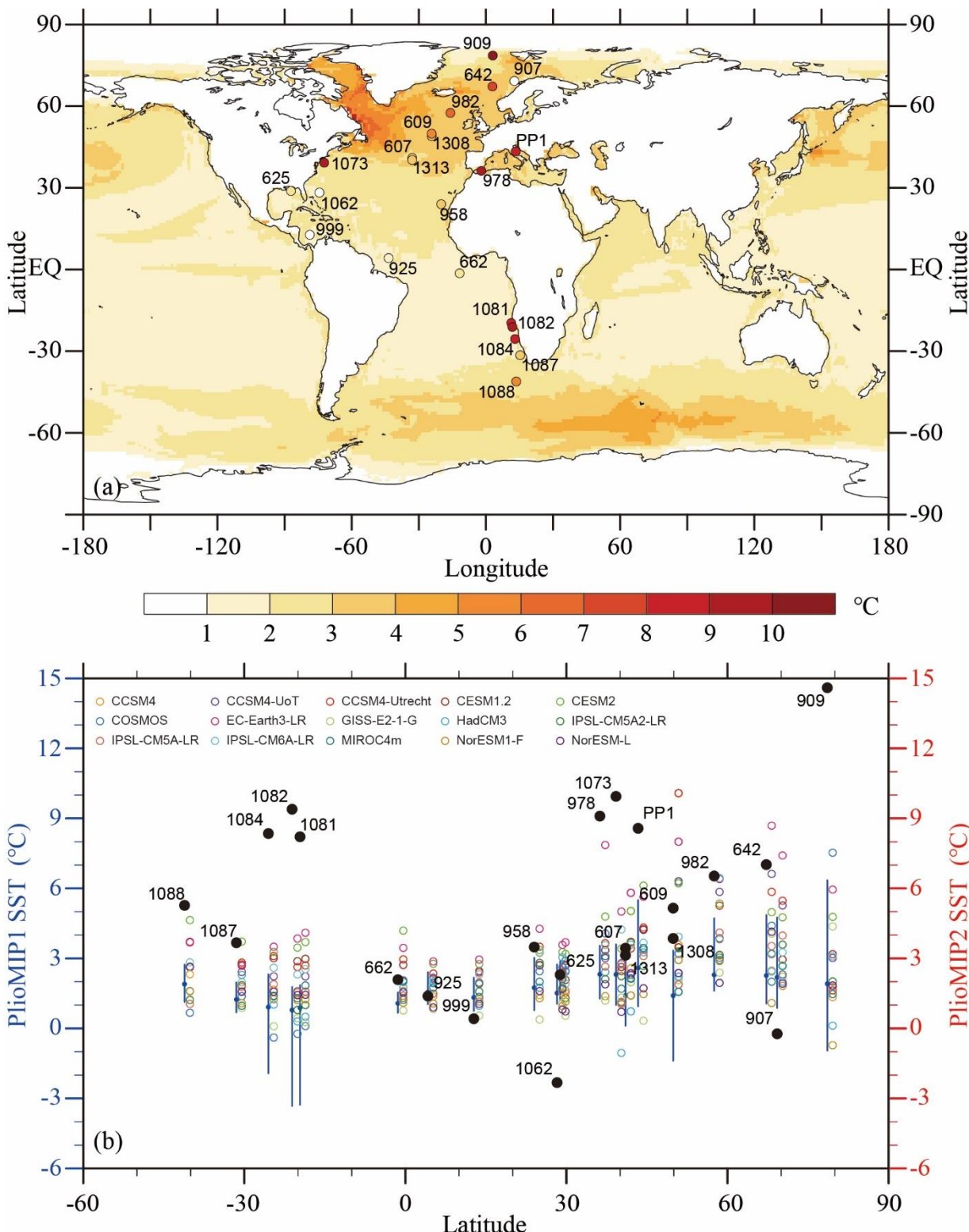

**Fig. 6. PlioMIP2 and PRISM4 SST comparison in the Atlantic.** (a) PRISM4 SST anomalies and data sites in the Atlantic and the Mediterranean, against with the multi-model ensemble median of SST anomalies (the mid-Pliocene vs. the pre-industrial) in PlioMIP2 (unit: °C). (b) Black dots show the PRISM4 SST anomalies (unit: °C). Vertical blue lines and dots show the PlioMIP1 ranges and median values of changes in SST for each site. Colored markers show SST changes simulated by each model in the PlioMIP2. The PRISM4 SST anomalies are calculated based on the PRISM4 mid-Pliocene reconstructions (3.19–3.22 Ma, Foley and Dowsett, 2019) and the modern observation (1870-1899, Rayner et al., 2003).

**Table 1. Comparison of PlioMIP2 models.**

| Model ID | Ocean resolution Lat. × Long. | Background vertical/diapycnal mixing | I. length/mean (years) | | Max AMOC | | | OHT* | OHT** | Refference |
|---|---|---|---|---|---|---|---|---|---|---|
| | | | PI | MP | PI | MP | (%) | (%) | (%) | |
| CCSM4 | 0.27–0.54°×1.1°, L60 depth | default KPP scheme[#]. k= 0.16 cm$^2$ s$^{-1}$ and latitudinally-varying | >1000/100 | 1100/100 | 26.6 | 29.6 | 11 | -7 | -6 | Feng et al., 2020 |
| CCSM4-UoT | 0.27–0.54°×1.1°, L60 depth | modified KPP scheme[$], identical k for PI and MP, k from 0.16×10$^{-4}$ to 1×10$^{-4}$ m$^2$ s$^{-1}$ and depth dependent | 4630/30 | 1250/30 | 22.6 | 23.5 | 4 | 9 | 9 | Chandan, et al., 2017,2018 |
| CCSM4-Utrecht | 0.27–0.54° ×1.1°, L60 depth | modified KPP scheme[$], uniform k= 0.16 cm$^2$ s$^{-1}$ for PI, but k from 0.1 to 1cm$^2$ s$^{-1}$ depth dependent for MP | 3100/100 | 2048/100 | 19.8 | 21.9 | 11 | 6 | 5 | Baatsen et al., 2020, in prep. |
| CESM1.2 | 0.27–0.54° ×1.1°, L60 depth | default KPP scheme | >1000/100 | 1200/100 | 26.7 | 27.0 | 1 | -10 | -9 | Feng et al., 2020 |
| CESM2 | 0.27–0.54° ×1.1°, L60 depth | default KPP scheme with Langmuir parameterization | 1200/100 | 1500/100 | 23.0 | 27.8 | 21 | -4 | -5 | Feng et al., 2020 |
| COSMOS | ~3.0° ×1.8°,L40 depth | k = 0.105 cm$^2$ s$^{-1}$ | 1950/100 | 1950/100 | 16.0 | 19.4 | 21 | 15 | 19 | Stepanek et al., 2020 |
| EC-Earth3-LR | 1.0° ×1.0°, L75 depth | k = 0.12 cm$^2$ s$^{-1}$ | 1500/100 | 1600/100 | 16.8 | 20.0 | 19 | 39 | 28 | Zhang et al., 2020 |
| GISS-E2-1-G | 1°×1.25°, L32 depth | KPP with nonlocal fluxes, k = 0.10 cm$^2$s$^{-1}$ | 5000/100 | 3100/100 | 28.2 | 35.1 | 24 | 4 | -1 | |
| HadCM3 | 1.25° ×1.25°, L20 depth | k = 0.10 cm$^2$ s$^{-1}$ | 2999/100 | 2499/100 | 15.4 | 20.7 | 34 | 38 | 30 | Hunter et al., 2019 |
| IPSL-CM5A2-LR | 0.5–2° ×2°, L31 depth | function of turbulent kinetic energy | 1500/100 | 3480/100 | 11.1 | 17.0 | 53 | 29 | 39 | Tan et al., 2020 |

| Model | Resolution | Mixing scheme | | | | | | | | | Reference |
|---|---|---|---|---|---|---|---|---|---|---|---|
| IPSL-CM5A-LR | 0.5–2° ×2°, L31 depth | function of turbulent kinetic energy | >800/100 | 3680/100 | 10.2 | 14.8 | 45 | 43 | 36 | Tan et al., 2020 |
| IPSL-CM6A-LR | 1.0° ×1.0°, refined at 1/3° in the tropics, L75 depth | turbulent kinetic energy scheme and an energy-constrained parameterization of mixing due to internal tides | 1100/100 | 1450/100 | 12.7 | 15.8 | 24 | 16 | 29 | Lurton et al., 2020 |
| MIROC4m | 0.56–1.4°×1.4°, L43 sigma/depth | k from 0.10 to 3 cm$^2$ s$^{-1}$, latitudinally varying | 2220/100 | 3000/100 | 19.6 | 20.2 | 3 | -10 | -10 | Chan et al., 2020 |
| NorESM1-F | ~1.0° ×1.0°, L53 sigma | k = 0.10 cm$^2$ s$^{-1}$, latitudinally varying | 2000/100 | 500/100 | 24.5 | 28.1 | 15 | 1 | 4 | Li et al., 2020 |
| NorESM-L | ~3.0° ×3.0°, L32 sigma | k = 0.10 cm$^2$ s$^{-1}$, latitudinally varying | 2200/100 | 1200/100 | 21.3 | 23.3 | 9 | -13 | -17 | Li et al., 2020 |

* North Atlantic ocean heat transport between 30°N and 80°N.

** Atlantic ocean heat transport between 30°S and 80°N.

# KPP (K-Profile Parameterization) scheme parameterizes boundary layer mixing and internal diabatic mixing by convection, shear instability, internal waves, tides, and double diffusion.

$ KPP parameterization, but with the overflow parameterization and the tidal mixing switched off

CESM2, EC-Earth3-LR, GISS-E2-1-G and IPSL-CM6A-LR take part in the Coupled Model Intercomparison Project (CMIP) phase 6.