# Peer review of "Mid-Pliocene Atlantic Meridional Overturning Circulation simulated in PlioMIP2"

_Climate of the Past, 2020_

## Referee Comment (RC1) · Chris Brierley (Referee) · 2 Nov 2020

I think this is a useful contribution to the discussion around the climate of the mid-Pliocene. Whilst this paper does not present a new discovery, it is a helpful description and preliminary of the results of a new batch of climate model simulations. It explores the impact of the some boundary condition changes to explain the difference between these new simulations and the previous ones. I have one comment about the scientific results, and then a series of comments about the manuscript text and presentation.

Science:

- In your discussion, you show how changes in the Arctic gateways are responsible for the higher AMOC strength seen in PlioMIP2 w.r.t. PlioMIP1. However, there

is no comment about how strong that impact is. My own work (Brierley Fedorov, 2016, https://www.sciencedirect.com/science/article/pii/S0012821X16300978) suggests that the closure of the Bering Strait results in an increase of just over 2 Sv, which is a similar value to that seen by Otto-Bliesner et al (2017). Stepanek et al (2020) get a value of just under 2 Sv. Yet you shy away from providing any estimates of the magnitude of this effect. I would like you to go a bit further.

- I also feel that your statement that the impact of the gateways is "highly model dependent" needs more explanation. I suspect you might be correct, but you have presented no evidence for in the manuscript and (as I've stated) the 2 papers you currently cite, as well as Brierley Fedorov (2016) get similar values for it.

Stylistic:

- There are too many lists throughout this manuscript. If you feel that it is so important to identify each individual model showing a behaviour, then please consider highlighting this in the table in some way.

- Please use the ESGF controlled vocabulary where possible. The HadGEM model should be HadGEM3-GC31-LL.

- The very long sentence ranging from L67-74 feels like it has been written to inflate certain metrics by including a large amount of self-citations. Please only cite work that is relevant to the topic in question, and be selective.

- L80. This SST value is a difference in a difference, and you haven't given the readers relevant context to assess the magnitude of these yet. [compared with last numbers on L75, they may think models simulate Pliocene cooling in region]. Please rephrase.

- L87. Remove "in 2016", as citation makes this clear.

- L89. Please give dates of KM5c

- L93-95. This sentence reads like you are only assessing over the subset of models that have run both PlioMIP1 and PiloMIP2. Is this really the case?

- L98. Compare -> investigate

- L100. models that participated

- L107-112. This sentence just repeats the information provided by the aforementioned table. Please remove it

- L126-128. If the only some of the 6 models have extended their piControl runs, how come there are new control simulations for the other models?

- Please be consistent between the names of CCSM4-Utrecht and CCSM4-UoT throughout the manuscript.

- L137. Is Kanzow et al (2011) really an observational estimate of the AMOC maximum?

- L139. Is it really fair to consider 1

- L151. If you test the significance of the 1

- L159: ranged -> ranging

- L171: Why have you chosen to plot the median here? I don't object to it, but there was no explanation for the choice.

- L175: Fig. 5 does not support this.

- L179: please remove this list and others (see first point)

- L190: Please change "underestimate" to "appear to underestimate". You are otherwise making an implicit statement about the source of error being in the models.

- L220: "the intensified" -> "an intensified"

- L223: I am not convinced about this statement. How can you exclude that the diffusivity has altered the preindustrial mean climate, which has then led to a different response through feedback processes?

- L231: carry -> carried

- L233-236: whilst I would agree with the statement, I am unsure how Hill et al and Feng et al evidence it.

- L239: Shouldn't you rather be looking at the median than mean, given Fig 6. Although I do wonder why Fig 6 is the median, and how that has not lead to discontuities.

- L254. The Benguela upwelling is not in the North Atlantic – why mention here?

- L265-266: Both Federov et al and Foley  Dowsett "reconstruct" SST not "show" it.

- L270-271. You have not mentioned the role of the overlying atmosphere resolution. This also matters – e.g. Gent et al (2010, Improvements in a half degree atmosphere/land version of the CCSM).

- L291. Why are you listing all 15 models here?

- Table 1. You may want to consider splitting this into 2 tables – a methods and results table.

[Figure]

- Table 1. Why does IPSL-CM5A have 2 lengths?

- Fig 6. PRSIM -> PRISM
* * *

---

## Referee Comment (RC2) · Anonymous Referee #2 · 13 Nov 2020

Review of "Mid-Pliocene Atlantic Meridional Overturning Circulation simulated in PlioMIP2" by Zhang et al. (2020)

The variation of AMOC in a warmer climate has always been the focus of climate research. The authors examined the response of AMOC and the associated changes in OHT and SST during the mid-Pliocene based on multi models in PlioMIP2. They reported a consistent increase in AMOC strength across models, whereas large model spread in terms of OHT and the depth of AMOC. More importantly, they highlighted that increased Pliocene AMOC and northward OHT maybe not a requisite for stronger warming over sub-polar North Atlantic, which is widely argued as the case in proxies. The manuscript is well written and organized and should be an important contribution to the understanding of the Pliocene climate.

[Figure]

While acceptable as it is, I think the following issue that deserves additional discussion to further improve the quality of the manuscript and may have a far-reaching impact beyond the scope of Pliocene. That is, why the increased AMOC is not accompanied by enhanced northward OHT or warming (e.g., in CCSM/CESM family), as this relationship, I think, was once considered to be "true" in most cases targeting paleoclimate. Is it possible that stronger AMOC transports more energy to the deep ocean (e.g., below mix layer?) instead of the surface?

Very minor comments: Line 53, add "over North Atlantic" after "warm climate"

Lines 70-73, investigation of polar ice sheets is also an important aspect of Pliocene climates, which should be discussed here.

Line 76, the pre-industrial->the pre-industrial level

Line 85, mid-Pliocene-> mPWP for consistency throughout the manuscript

Line 87, add "the" before "state-of-the-art"

Line 101, participated in-> participating

Line 136, why use the median value instead of the multimodel ensemble mean? Please discuss

Line 142, how to measure the depth of AMOC in models? The mean depth of the AMOC cell (e.g., positive stream function?). Please define.

Line 205, considers-> consider

Line 206, participated in-> participated

―――――――――――――――――――――

---

## Author Comment (AC1) · 13 Nov 2020

Since the modelling groups in the PlioMIP2 do not carry out consistent sensitivity experiments for the Arctic gateways, it remains difficult to compare the impacts of the gateways on AMOC. However, some sensitivity experiments done in several groups do show a clear model-spread. Here, we summarize some results from these sensitivity experiments.

1) Sensitivity experiments to the Bering Strait

With the PlioMIP1 boundary conditions, CCSM4 has carried out a sensitivity experiment for the Bering Strait (Otto-Bliesner et al., 2017). The closed Bering Strait leads to an AMOC strengthened by $\sim$2.5 Sv ($\sim$10%).

Based on the PlioMIP1 boundary conditions, COSMOS simulates that the closed Bering Strait makes the AMOC increased by ∼1.8 Sv (∼11%).

MIROC4m has done a sensitivity experiment for the Bering Strait, but based on the pre-industrial boundary conditions. With an open and closed Bering Strait, the AMOC maximum is 19.6 and 21.65 Sv, respectively. Thus, closing the strait in the pre-industrial increases the AMOC by ∼2.0 Sv (∼10%).

In addition to the Eoi400 experiment, CCSM-UoT has done a sensitivity simulation with the Bering Strait opened. Model results show that the closed strait causes the AMOC enhanced by ∼4.2 Sv (∼25%).

2) Sensitivity experiments to the Arctic gateways (the Bering Strait and the Canadian Arctic Archipelago)

With the PlioMIP1 boundary conditions, CCSM4 has done a sensitivity experiment with the Bering Strait and the Canadian Arctic Archipelago closed. The model responds with an even greater strengthening of the AMOC (∼4.5 Sv or ∼18%), approximately doubling the response with only the Bering Strait closed (Otto-Bliesner et al., 2017).

Based on the PlioMIP2 boundary conditions, COSMOS has run a sensitivity experiment with the Bering Strait, the Hudson Bay and the Canadian Arctic Archipelago opened. The comparison between the Eoi400 and this experiment shows that the closing of the gateways only leads to an enhancement in AMOC of 1.68 Sv (∼9%).

These results show the model-spread in simulating AMOC responses to the modification of the Arctic gateways. Although CCSM-UoT shares many similarities with CCSM4, CCSM-UoT produces a much large response in AMOC than CCSM4. Moreover, CCSM4 suggests that the Canadian Arctic Archipelago is also an important factor that influences the intensity of AMOC, whereas COSMOS shows that its impact seems small.

---

## Editor Comment (EC1) · Appy Sluijs (Editor) · 24 Nov 2020

Dear authors,

Even though the discussion phase has passed, I still very much hope to receive your reply to the issues (s)he raised.

All the best,

Appy Sluijs Editor

---

## Author Comment (AC2) · 24 Nov 2020

Dear Editor and the reviewer,

Sorry for the delay. After the discussion within the PlioMIP2 group, we reply this question now.

Although an enhanced AMOC can warm high-latitudes up, observations of strong high-latitude warming in the North Atlantic are not sufficient to constrain the strength of the AMOC or OHT (Zhang et al., 2013).

The strength of AMOC measures the V contrast between upper and lower branch of the Atlantic transport, but the OHT is also influenced by the T contrast as well as the depth of AMOC. Moreover, OHT can be decomposed into a (vertical) MOC component

and a (horizontal) gyre component. While the MOC component dominates in most of the Atlantic region, the gyre component has a comparable magnitude in the subpolar region (Williams et al., 2015). Therefore, there is not a one-to-one correspondence between MOC and OHT, especially in the subpolar regions.

Furthermore, the SST warming pattern is not entirely determined by OHT. The PlioMIP models show this effect prominently, although there is probably some warming due to the simulated enhanced OHT. As suggested by an early study (Feng et al., 2017), enhanced warming in the Northern high latitudes is primarily a result of regional feed-backs.

Regards

Zhongshi Zhang and all coauthors

Zhang, Z., Nisancioglu, K.H., and Ninnemann, U.S.: Increased ventilation of Antarctic deep water during the warm mid-Pliocene. Nature Communications. 4, 1499, 2013.

Williams, R.G., Roussenov, V., Lozier, M.S., Smith, D.: Mechanisms of Heat Content and Thermocline Change in the Subtropical and Subpolar North Atlantic. Journal of Climate, 28 (24), 9803–9815, 2015.

Feng, R., Otto-Bliesner, B. L., Fletcher, T. L., Tabor, C. R., Ballantyne, A. P., & Brady, E. C.: Amplified Late Pliocene terrestrial warmth in northern high latitudes from greater radiative forcing and closed Arctic Ocean gateways. Earth and Planetary Science Letters, 466, 129-138, 2017.

―――――――――――――

---

## Author Response (AR1)

Dear Editors and Reviewers,

Thank you very much for the constructive comments and suggestions. We have taken all these comments into account in the revised version. All revisions are highlighted in red in the revised paper.

Our detailed answers are listed below. All reviewers' comments are shown in blue, and our answers are shown in black.

Zhongshi Zhang and all coauthors

Reviewer 1
I think this is a useful contribution to the discussion around the climate of the mid-Pliocene. Whilst this paper does not present a new discovery, it is a helpful description and preliminary of the results of a new batch of climate model simulations. It explores the impact of the some boundary condition changes to explain the difference between these new simulations and the previous ones. I have one comment about the scientific results, and then a series of comments about the manuscript text and presentation.

Science:
• In your discussion, you show how changes in the Arctic gateways are responsible for the higher AMOC strength seen in PlioMIP2 w.r.t. PlioMIP1. However, there is no comment about how strong that impact is. My own work (Brierley Fedorov,2016,https://www.sciencedirect.com/science/article/pii/S0012821X16300978) suggests that the closure of the Bering Strait results in an increase of just over 2Sv, which is a similar value to that seen by Otto-Bliesner et al (2017). Stepanek et al (2020) get a value of just under 2 Sv. Yet you shy away from providing any estimates of the magnitude of this effect. I would like you to go a bit further.

• I also feel that your statement that the impact of the gateways is "highly model dependent" needs more explanation. I suspect you might be correct, but you have presented no evidence for in the manuscript and (as I've stated) the 2 papers you currently cite, as well as Brierley Fedorov (2016) get similar values for it.

In the revised version, we add two sentences in line 211 to 213. "Some simulations suggest that the AMOC is enhanced by ~2 Sv due to the closed Bering Strait (Brierley and Fedorov, 2016; Otto-Bliesner et al., 2017), while some unpublished simulations in PlioMIP2 show much larger responses. Without consistent sensitivity experiments for the Arctic gateways, it remains difficult to reveal the range of model-spread on the gateways' impacts in PlioMIP2. This model-dependence will be addressed in more dedicated sensitivity experiments in the future."

Here, since gateway sensitivity experiments have not been published yet by some modelling groups, we do not include the values in AMOC responses in the revised version, but these values can be found in the discussion.

Stylistic:
• There are too many lists throughout this manuscript. If you feel that it is so important to identify each individual model showing a behaviour, then please consider highlighting this in the table in some way.
In the revised version, we have removed some of the lists.

• Please use the ESGF controlled vocabulary where possible. The HadGEM modelshould be HadGEM3-GC31-LL.
Done

• The very long sentence ranging from L67-74 feels like it has been written to inflate certain metrics by including a large amount of self-citations. Please only cite work that is relevant to the topic in question, and be selective.
We delete this long sentence (as well as the reference list) in the revised version.

• L80. This SST value is a difference in a difference, and you haven't given the readers relevant context to assess the magnitude of these yet. [compared with last numbers on L75, they may think models simulate Pliocene cooling in region].Please rephrase.
The sentence is rewritten in line 76. "The PlioMIP1 simulated surface warming in the North Atlantic is ~4–6°C smaller than the reconstruction."

• L87. Remove "in 2016", as citation makes this clear.
Done

• L89. Please give dates of KM5c
Done

• L93-95. This sentence reads like you are only assessing over the subset of models that have run both PlioMIP1 and PiloMIP2. Is this really the case?
The sentence is rewritten. See line 88.

• L98. Compare -> investigate
Done

• L100. models that participated
Done

• L107-112. This sentence just repeats the information provided by the aforementioned table. Please remove it

Done. See line 101

• L126-128. If the only some of the 6 models have extended their piControl runs,how come there are new control simulations for the other models?
The sentence is reworded. See line 119.

• Please be consistent between the names of CCSM4-Utrecht and CCSM4-UoT throughout the manuscript.
Done

• L137. Is Kanzow et al (2011) really an observational estimate of the AMOC maximum?
The sentence is changed. "The multi-model median value of the AMOC maximums is 19.8 Sv, which is comparable to the observational AMOC strength of $18.7 \pm 2.1$ Sv at 26.5°N (Kanzow et al. 2010)"

• L139. Is it really fair to consider 1. • L151. If you test the significance of the 1
Since significant tests are not used in this study, the changes of 1 still need to be listed, though they are small and might be insignificant. Furthermore, even these small changes are insignificant, they are not misleading but properly show the large range of model-spread. In the revised version, we add a few sentences in figure captions to explain that significant tests are not used.

• L159: ranged -> ranging
Done.

• L171: Why have you chosen to plot the median here? I don't object to it, but there was no explanation for the choice.
In the revised version line 115, we explain this point. "Note five models come from the CCSM/CESM family in the PlioMIP2 ensemble. To avoid these models taking undue weights in the PlioMIP2 ensemble, median instead of mean values are used in this study."

• L175: Fig. 5 does not support this.
Fig 5 is deleted in the bracket.

• L179: please remove this list and others (see first point)
Done

• L190: Please change "underestimate" to "appear to underestimate". You are otherwise making an implicit statement about the source of error being in the models.
Done. Also revised in line 51 and 309.

• L220: "the intensified" -> "an intensified"

Done

• L223: I am not convinced about this statement. How can you exclude that the diffusivity has altered the preindustrial mean climate, which has then led to a different response through feedback processes?

CCSM4-Utrecht uses the default k in the PI control run, but the enhanced k in the mid-Pliocene experiment. In the revised version, we cite the table 1 to show the difference in k values

• L231: carry -> carried

Done

• L233-236: whilst I would agree with the statement, I am unsure how Hill et al and Feng et al evidence it.

The sentence is rewritten line 233-239. "Earlier studies (e.g., Feng et al., 2017) have noticed that the North Atlantic warming is not a unique feature in many mid-Pliocene simulations, since the warming in the North Pacific is also remarkable (Fig 4). This inter-basin symmetry suggests a potentially important component of the zonal mean polar amplification of the SST warming across the North Atlantic. Energy balance analyses (Hill, 2014; Feng et al., 2017) show that amplified zonal mean northern high latitude warming is dominated by regional radiative feedbacks from lowered surface albedo and enhanced high latitude greenhouse effect (from changes in water vapor), even with an enhanced AMOC by gateway closure."

• L239: Shouldn't you rather be looking at the median than mean, given Fig 6. Although I do wonder why Fig 6 is the median, and how that has not lead to discontuities.

Please see our reply above.

• L254. The Benguela upwelling is not in the North Atlantic – why mention here?

Changed to "Although the model-data discrepancy is reduced in the North Atlantic partly due to the intensified AMOC, the model-data mismatch remains large in other regions in PlioMIP2, for example Sites 1081, 1082, 1084, 1087 in the Benguela upwelling region (Fig. 6)."

• L265-266: Both Federov et al and Foley Dowsett "reconstruct" SST not "show"it.

Done.

• L270-271. You have not mentioned the role of the overlying atmosphere resolu-tion. This also matters – e.g. Gent et al (2010, Improvements in a half degree atmosphere/land version of the CCSM).

Added.

• L291. Why are you listing all 15 models here?• Table 1. You may want to consider

splitting this into 2 tables – a methods and results table.
The list is deleted. However, the Table 1 is not split.

Table 1. Why does IPSL-CM5A have 2 lengths?
The length is revised.

• Fig 6. PRSIM -> PRISM
Done.

Reviewer 2
The variation of AMOC in a warmer climate has always been the focus of climate research. The authors examined the response of AMOC and the associated changes in OHT and SST during the mid-Pliocene based on multi models in PlioMIP2. They reported a consistent increase in AMOC strength across models, whereas large model spread in terms of OHT and the depth of AMOC. More importantly, they highlighted that increased Pliocene AMOC and northward OHT maybe not a requisite for stronger warming over sub-polar North Atlantic, which is widely argued as the case in proxies. The manuscript is well written and organized and should be an important contribution to the understanding of the Pliocene climate.

While acceptable as it is, I think the following issue that deserves additional discussion to further improve the quality of the manuscript and may have a far-reaching impact be-yond the scope of Pliocene. That is, why the increased AMOC is not accompanied by enhanced northward OHT or warming (e.g., in CCSM/CESM family), as this relation-ship, I think, was once considered to be "true" in most cases targeting paleoclimate. Is it possible that stronger AMOC transports more energy to the deep ocean (e.g., below mix layer?) instead of the surface?

We add a new paragraph to explain this point in the revised version. Please see line 230-238.

Very minor comments:
Line 53, add "over North Atlantic" after "warm climate"
Done.

Lines 70-73, investigation of polar ice sheets is also an important aspect of Pliocene climates, which should be discussed here.
This sentence is removed according to the comment from Reviewer 1.

Line 76, the pre-industrial->the pre-industrial level
Done.

Line 85, mid-Pliocene-> mPWP for consistency throughout the manuscript

Since PlioMIP focuses on the KM5c interglacial period, it better not to use mPWP throughout the manuscript. However, in this sentence the mPWP is used in the revised version. See line 84.

Line 87, add "the" before "state-of-the-art"
Done.

Line 101, participated in-> participating
Change to "the models that participated in PlioMIP2".

Line 136, why use the median value instead of the multimodel ensemble mean? Please discuss.
In the revised version line 115, we explain this point. "Note five models come from the CCSM/CESM family in the PlioMIP2 ensemble. To avoid these models taking undue weights in the PlioMIP2 ensemble, median instead of mean values are used in this study."

Line 142, how to measure the depth of AMOC in models? The mean depth of theAMOC cell (e.g., positive stream function?). Please define.
Done.

Line 205, considers-> consider
Done.

Line 206, participated in-> participated
Done.